# N Addition Mitigates Water Stress via Different Photosynthesis and Water Traits for Three Native Plant Species in the Qinghai–Tibet Plateau

**Ningning Zhao** [1,2,3], **Xingrong Sun** [3,4], **Shuai Hou** [3,4], **Guohao Chen** [1], **He Zhang** [2,3], **Yuxin Han** [3,4], **Jie Zhou** [3,4], **Xiangtao Wang** [3,4,*] **and Zhixin Zhang** [1,*]

1  College of Grassland Agriculture, Northwest A&F University, Yangling 712100, China
2  College of Resources and Environment, Tibet Agriculture and Animal Husbandry University, Nyingchi 860000, China
3  Qiangtang Alpine Grassland Ecosystem Research Station (Jointly Built with Lanzhou University), Tibet Agricultural and Animal Husbandry University, Nyingchi 860000, China
4  College of Animal Science, Tibet Agriculture and Animal Husbandry University, Nyingchi 860000, China
*  Correspondence: wangxt@xza.edu.cn (X.W.); zhixin@nwafu.edu.cn (Z.Z.)

**Abstract:** Reseeding with native plants to rebuild alpine meadow has become a popular way of ecological restoration. However, the harsh environment poses a great challenge to the establishment of native plants due to poor management of water and nutrients. How water–fertilizer interaction influences dominant grass species is still unclear, and reasonable water and fertilizer conditions are still not determined. Our results showed that addition of nitrogen could mitigate the photosynthetic and water-use traits caused by water stress, i.e., a reduction in $P_n$ and water use results from fewer and thinner leaves, weak stomatal traits, etc. Compared to the control, the peak $P_n$ values of *Poa crymophila*, *Festuca coelestis*, and *Stipa purpurea* increased significantly (71.2%, 108.4%, and 25.4%, respectively). Under drought stress, $P_n$ tended to decrease due to reduced stomatal conductance ($G_s$). However, appropriate fertilization buffered against $P_n$ decreases by altering the stomatal size and regulating the $G_s$. Based on reduced water consumption, the water-use efficiency of *P. crymophila* and *F. coelestis* decreased whereas that of *S. purpurea* increased. $W_H F_H$ for *P. crymophila* and *F. coelestis* and $W_H F_L$ for *S. purpurea* growth were suitable for the alpine region. $W_H F_H$ for *P. crymophila* and *F. coelestis* and $W_H F_L$ for *S. purpurea* were suitable for their establishment in the alpine region. A reasonable water–fertilizer combination could effectively reduce the risk of establishment failure in ecological restoration.

**Keywords:** *Poa crymophila*; *Festuca coelestis*; *Stipa purpurea*; stomatal characteristic; water-use efficiency; alpine meadow; ecological restoration

## 1. Introduction

The Qinghai–Tibet Plateau is the highest (average elevation of 4000 m) and largest plateau in the world [1]. The ecosystem is fragile and sensitive, and over the past three decades, the alpine grasslands have witnessed an increasing trend in temperature and the influence of climate change in combination with an uneven rainfall within the growing season and stress from overgrazing, resulting the suffering of a large area of grassland from drought stress and degradation [2–4]. These disturbances are challenging the sustainability of the Qinghai−Tibet Plateau's grassland ecosystems [5]. Therefore, there is an urgent need for appropriate ecological restoration methods. Reseeding is a key method for restoring degraded grasslands [6]. The results of previous studies have shown that the degradation could be effectively mitigated via the planting of native plants due to their adaptability advantages [7]. However, the ratio of success is limited by the large variation in rainfall in degraded areas, which poses a further challenge for drought and nutrient management.

The absorption and use of nutrients of plants is closely related to the soil's water content. Therefore, the soil moisture largely determines the effectiveness of fertilizers, and

also becomes an important factor for plant growth and development [8]. The stomata is the main site for water and $CO_2$ exchange between plant leaves and the external environment and it responds rapidly to water conditions. Drought stress can cause stomatal closure and may even damage mesophyll cells, which reduces the activity of photosynthetic enzymes and consequently retards photosynthesis [9,10]. Plants also respond to drought by regulating their physiological characteristics such as growth rates, cell osmotic potentials, and antioxidant defense systems [11]. Fertilizer application is a direct method for supplementing the vegetation of degraded grasslands with nutrients. The application of nitrogen fertilizer promotes plant root growth, which increases water absorption and in turn promotes photosynthesis and plant growth, thereby mitigating the effects of water deficiency. However, excessive nitrogen fertilizer application reduces photosynthesis and produces soil pollution issues [12–16]. Previous research has demonstrated that insufficient or excessive water and fertilizer inhibit plant growth and development [17]. Therefore, optimizing the water−fertilizer relationship to harness their combined advantages is essential for the success of reseeding and restoration of degraded grasslands.

Distributed over the northeastern parts of the Qinghai−Tibet Plateau is one of the largest areas of alpine meadow in China. Here, grassland degradation and reductions in the soil nutrient and soil water contents are particularly severe, caused by the poor management and overuse of grassland [18]. Native plants have a verified merit and potential for use in ecological restoration. A preliminary study on reseeding and fertilizer application demonstrated that different reseeding combinations with *Melissitus rutenica*, *Puccinellia tenuiflora*, *Elymus nutans*, *Stipa aliena*, and *Koeleria cristata* increased the community stability [19]. Fertilizer application and reseeding with *Poa pratensis* in a striped pattern effectively enhanced the community diversity and functional structure in an alpine meadow [20]. Previous research also revealed that using *Kobresia* species, *Elymus nutans*, *Bromus inermis* Layss, and *P. crymophila* as native dominant plants for reseeding clearly alleviated grassland degradation, which provided considerable economic and ecological value [21,22]. However, the adaptive capabilities of the major native dominant plant species under different environments still remains unclear.

*Poa crymophila*, *Festuca coelestis*, and *Stipa purpurea* are dominant species native to the alpine meadow, belong to perennial gramineous grasses, and have high adaptability in drought, low-temperature, strong radiation, and nutrient-deficient environments. They are important food sources for ruminants and very good candidates for ecological restoration. Currently, there are only a few studies on *F. coelestis*, and studies on *P. crymophila* and *S. purpurea* have been limited to the investigation of the effects of a single fertilizer [23,24] or drought [25]. Further exploration is required to determine the conditions and potential of these species for supplementary sowing applications in degraded grasslands and to elucidate the effects of water−fertilizer coupling combinations and the mechanisms of adaptation. Therefore, this study with a water- and fertilizer-controlled pot experiment was conducted to: (1) determine the photosynthetic response of *P. crymophila*, *F. coelestis*, and *S. purpurea* under different water−fertilizer treatments; (2) analyze the water−fertilizer coupling interactions in water use and dry matter production; and (3) determine the optimum water−fertilizer combination of the test plant species. Our results are conducive to the sustainability of alpine grassland ecosystems and provides new options for degraded grassland restoration.

## 2. Materials and Methods

### 2.1. Study Area and Experimental Design

A pot experiment was conducted at the Grassland Science Internship Base of the Tibet Agricultural and Animal Husbandry University located in the Bayi Subdistrict of Bayi District of Nyingchi City of the Tibet Autonomous Region (29.66° N, 94.34° E; 2969.64 m above sea level). The study site is characterized by a plateau temperate humid/semi-humid monsoon climate, with the peak phase of the rainy season generally occurring between the end of June and the end of August. The study area has an average annual temperature

of 7−16 °C, an average annual precipitation of 650 mm, an annual total solar radiation of 5460−7530 MJ/m$^2$, and an average annual relative humidity of 71%.

The major gramineous forage grass species of the North Tibetan alpine grassland, namely *P. crymophila*, *F. coelestis*, and *S. purpurea*, were selected for this experiment. Grass seeds were collected in September 2020 and stored at 5 °C before initiation of the experiment. A controlled pot experiment was conducted from August to December 2021. Cylindrical pots with a height of 20 cm and an internal diameter of 28.5 cm were used for the experiment. The pots were filled with sandy loam with a pH of 7.30, 25.73 mg·kg$^{-1}$ of fast-release nitrogen fertilizer, and 9.30 mg·kg$^{-1}$ of fast-release phosphorus fertilizer. Each pot was filled with 8.5 kg of sandy loam. The field capacity (FC) and bulk density of the soil were 30.27% and 1.32 g/cm$^3$, respectively. A two-factor split-plot experimental design was employed, with water and nitrogen fertilizer treatment serving as the two factors. Three water level treatments were set, namely soil water contents of 75% (W$_H$, adequate water), 55% (W$_M$, mild water deficiency stress), and 35% (W$_L$, moderate water deficiency stress). In terms of nitrogen fertilizer treatment, urea [CO(NH$_2$)$_2$] was used as the nitrogen fertilizer and four treatment levels were set, e.g., control (0 g/kg, CK), 0.11 g/kg (F$_L$), 0.33 g/kg (F$_M$), and 0.54 g/kg (F$_H$). A control group was established with 0 g/kg of fertilizer treatment. Thirty-six different water−fertilizer treatments were used and six replicates were established for each treatment (for a total of 216 pots).

Sowing was performed on 6 August 2021, with ten seeds evenly sowed at the center of each pot. After the seedlings emerged, five seedlings with similar growth states were retained in each pot. On 6 August 2021, phosphorus pentoxide (0.07 g/kg) was applied to all pots for each treatment group. Potassium chloride (0.1 g/kg) was used as the base fertilizer, and the pots of all treatment groups were watered to achieve a soil water content of 60%. One week after applying the base fertilizer, urea was added to the various treatment groups. The amount of fertilizer applied to each pot was converted to dose per unit area, and urea supplementation was only performed once. From this point onwards (until the end of the experiment), the soil water contents of the various treatment groups were strictly controlled using the gravimetric method. The pot positions of the various treatment groups were changed once weekly to ensure the consistency of the experiment.

### 2.2. Measured Indicators and Measurement Methods

#### 2.2.1. Photosynthetic Characteristics and Water−Use Efficiency (WUE) of Plants

Photosynthetic characteristics were measured once every 2 h between 9:00 and 19:00 on a sunny day (7 December 2021) using an LI−6400 portable photosynthesis system (LI−COR Corporation, Lincoln, NE, USA). Three replicates were randomly selected for each treatment, and two representative leaf blades were selected from each replicate to measure the net photosynthetic rate (P$_n$), transpiration rate (T$_r$), stomatal conductance (G$_s$), and WUE at approximately halfway up the leaf to the leaf tip. Three repeat measurements were taken for each indicator, and the average values were recorded as the measured values. The light intensity of the photosynthesis system was set as 800 μmol·m$^{-2}$·s$^{-1}$. The CO$_2$ concentration within the leaf chamber was set as 350 μmol/mol, and the CO$_2$−flow rates were set as 300 μmol·s$^{-1}$ for *P. crymophila* or 250 μmol·s$^{-1}$ for *F. coelestis* and *S. purpurea*.

#### 2.2.2. Leaf Blade Stomatal Density and Size Measurements

The lower epidermis of leaf blades of the experimental plants was obtained to prepare temporary slices, which were placed under a microscope at high magnification (40×) to observe the leaf blade stomata. For each slice, 10 fields of view (FOVs) were randomly selected and photographed. The stomatal density was calculated using AutoCAD 2021 software (Autodesk Corporation, San Francisco, CA, USA) based on the microscopic magnification using the following formula: stomatal density = number of stomata in the FOV/area of the FOV.

### 2.2.3. Plant Traits

Measurements were performed with the leaf blades of all three plant species. Three replicates were obtained for each treatment group, and two plants were randomly selected for each replicate.

Two plants were selected from each pot to determine the leaf blade counts. Two leaf blades were selected from each plant to measure the leaf blade thicknesses at approximately halfway up the leaf tip using a Vernier caliper.

### 2.2.4. Water-Use Efficiency of Plants

Sample destruction was performed at the end of the experiment. Soil layer samples were obtained at 10 cm depth intervals (two layers in total), and three replicate samples were obtained for each treatment. The samples were dried in an oven at 65 °C for 48 h before measuring the gravimetric water content of the soil. The FC was measured using the ring knife method and calculated using the formula for soil water storage (SWS) [26]:

$$SWS = D \times H \times W \times 10$$

where D, H, and W represent the bulk density (g/cm$^3$), depth (cm), and gravimetric water content (%) of the soil, respectively.

Water consumption was calculated using the following formula [27]:

$$WU = SWS_1 - SWS_2 + I$$

where $SWS_1$ and $SWS_2$ represent the SWS (mm) during the sowing and harvesting stages, respectively, and I represents the irrigation level. Given that the pot experiment was conducted beneath a rain shelter, it was assumed that the water consumption by the crops mainly reflected SWS and irrigation, and other sources of water were not considered.

### 2.2.5. Plant Biomass

One plant was randomly selected from the replicates of each treatment group. Extraneous matter was removed by washing with water, and the aboveground and belowground parts of the plants were separated. Samples were dried in an oven at 65 °C until a constant weight was attained, and the plants were subsequently weighed.

### *2.3. Statistical Analysis*

The measured data were consolidated using Microsoft Excel 2003. Univariate and bivariate significance testing was performed with SPSS (version 21.0, IBM Corp, Armonk, NY, USA) at the $\alpha = 0.05$ significance level, and multiple comparison testing was performed using the least significant difference test. The results of our statistical analyses were plotted using Origin 2021 software (OriginLab Corporation, Northampton, MA, USA).

## 3. Results

Significant differences (* $p < 0.05$) were observed in terms of the effects of water, fertilizer, and their interactions on the $P_n$ values of *P. crymophila*, *F. coelestis*, and *S. purpurea*.

The $P_n$ values of *P. crymophila* and *F. coelestis* tended to exhibit double or single peaks under different water–fertilizer conditions, whereas the $P_n$ values of *S. purpurea* exhibited double peaks under all water–fertilizer conditions. In *P. crymophila*, under the $W_H$ and $W_L$ conditions, the peak $P_n$ values attained with $F_H$ treatment occurred from 11:00−13:00 and 15:00−17:00, respectively. The peak $P_n$ value under the $W_H F_H$ treatment was significantly higher than that under other treatments and 71.2% higher than that under the CK. Under the $W_M$ treatment, the amount of fertilizer applied had a smaller effect on $P_n$. For *F. coelestis*, the $P_n$ exhibited a double-peak pattern under the $W_H$ and $W_M$ treatments, with peak values occurring from 11:00−13:00 and 15:00−17:00, respectively. Under the $W_H F_H$ treatment, the $P_n$ value was higher than under the other treatments, with the increase being as high as 108.4% when compared with that in the CK. Under the $W_L$ condition, the $P_n$ value

exhibited a single peak that occurred from 11:00–13:00. In *S. purpurea*, the highest $P_n$ peak values occurred from 11:00−13:00 and 15:00−17:00 under the $W_H F_L$ treatment, which is 41.3% higher than that observed under the CK.

Under different water content levels, differences were found in the responses of the daily average $P_n$ to fertilizer applications. In *P. crymophila*, the main differences occurred under the $W_H$ and $W_L$ conditions, with the daily average $P_n$ values under the $F_H$ treatment being 33.1% and 21.8% higher than that under the CK condition, respectively. The effects of the water content level also differed when different fertilizer levels were applied. For example, the water content levels under the CK was significantly different ($p < 0.05$) from those under the $W_M$ and $W_L$ treatments; furthermore, the water content levels were significantly different ($p < 0.05$) under the $F_M$ and $F_H$ treatments. For *F. coelestis*, the daily average $P_n$ value increased with increasing fertilizer application levels. Under the $W_H$ condition, differences in the daily average $P_n$ value existed across the various fertilizer application levels, with the difference between the $F_H$ and CK groups being the greatest (66.2% higher than that of the CK). Under the $W_M$ and $W_L$ conditions, the daily average $P_n$ values under the $F_H$ and $F_M$ treatments increased by 56.7% and 58.5% and by 35.6% and 28.0%, respectively, when compared with those of the CK. Under the CK condition, differences in daily average $P_n$ value between the $W_L$, $W_H$, and $W_M$ treatments were statistically significant; under the $F_L$ treatment, the differences in the daily average $P_n$ values between $W_M$ treatment and the other two water treatments were significant ($p < 0.05$). In *S. purpurea* under the $W_H$ condition, the daily average $P_n$ value was highest under the $F_L$ treatment. The daily average $P_n$ value under the $W_L F_M$ treatment was 25.4% higher than that under the CK; under the $F_M$ condition, the daily average $P_n$ values decreased in the following order of water content levels: $W_L > W_M > W_H$. Under other fertilizer treatments, the daily average $P_n$ decreased as water content decreased (Figure 1).

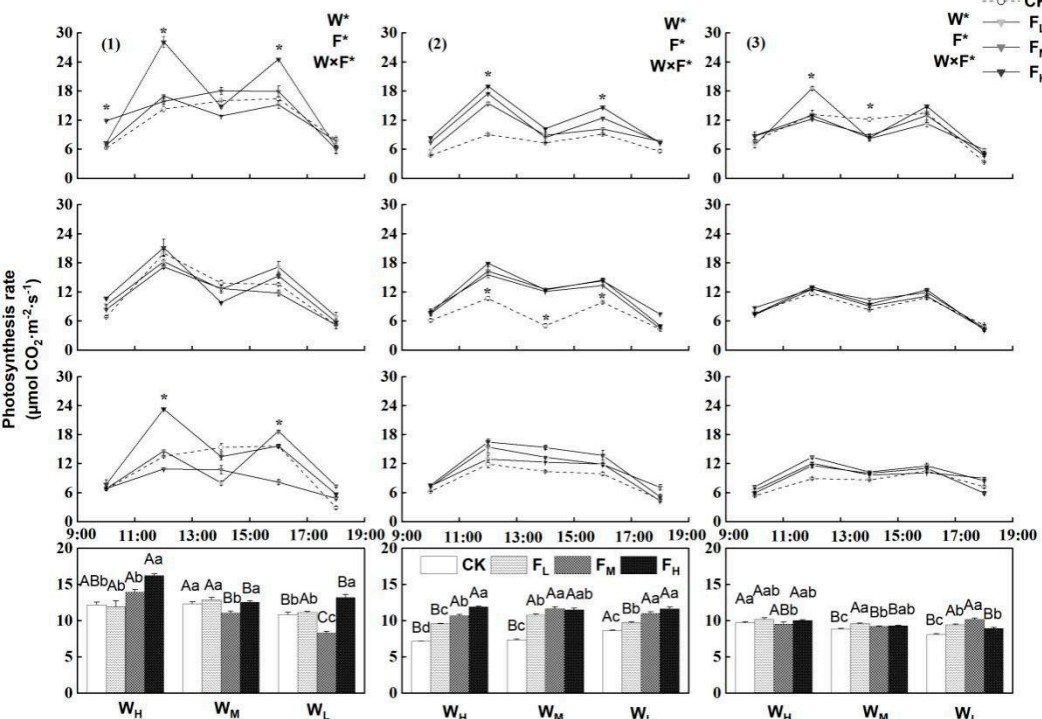

**Figure 1.** Changes in $P_n$ values of test plant species under different water–fertilizer treatments. Numbers (**1**)−(**3**) in brackets indicate *P. crymophila*, *F. coelestis*, and *S. purpurea*, respectively. W, F, and W×F indicate the results of ANOVAs ($p < 0.05$) for the effects of water, fertilizer, and their interactions. Asterisk (*) and ns indicate significant difference and non-significant. The uppercase letters and lowercase letters represent the difference between water and fertilizer treatments.

The effects of water, fertilizer, and their interactions on leaf number and leaf thickness differed significantly among the *P. crymophila* ($p < 0.05$). In *F. coelestis*, water, fertilizer, and their interactions caused significant differences in leaf thickness, whereas only fertilizer caused significant differences in leaf number. In *S. purpurea*, only water and fertilizer caused significant differences in leaf number, whereas all factors except water caused significant differences in leaf thickness ($p < 0.05$).

The leaf number and leaf thickness of all three species decreased with an increase in fertilizer application level. Leaf number was highest under the $W_H F_L$ treatment for all three species, with the values being 50.7, 37.2, and 24.3, which were increased by 198.1%, 43.9%, and 39.0%, respectively, compared those of the CK. At the same fertilizer application level, the leaf numbers of all three species decreased with the reduction in water content. Leaf thickness of *P. crymophila* and *F. coelestis* decreased gradually with a decrease in water content level, whereas leaf thickness of *S. purpurea* was greatest under the $W_H F_L$ treatment, which increased by 115.4% compared with the control group ($p < 0.05$; Figure 2).

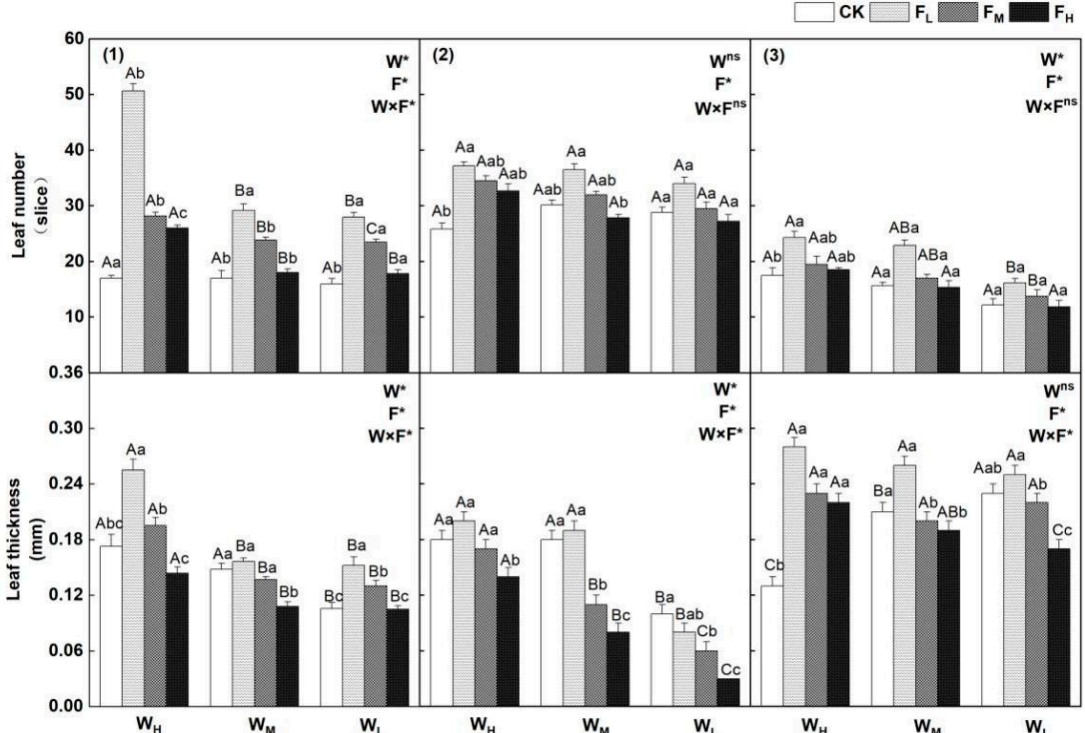

**Figure 2.** Changes in leaf number and leaf thickness values of test plant species under different water–fertilizer treatments. Numbers (**1**)−(**3**) in brackets indicate *P. crymophila*, *F. coelestis*, and *S. purpurea*, respectively. W, F, and W×F indicate the results of ANOVAs ($p < 0.05$) for the effects of water, fertilizer, and their interactions. Asterisk (*) and ns indicate significant difference and non-significant. The uppercase letters and lowercase letters represent the difference between water and fertilizer treatments.

Water, fertilizer, and their interactions exerted significant effects on the aboveground biomass for all three species ($p < 0.05$).

The highest under the $W_H F_L$ treatment for the three species, with the values for *P. crymophila*, *F. coelestis*, and *S. purpurea* being 0.2232, 0.1991, and 0.1327, respectively. For *P. crymophila*, under the $W_H$ and $W_L$ conditions, the aboveground biomass with the $F_L$ treatment increased by 137.1% and 273.1%, when compared with those under the CK. Under the $W_M$ condition, the aboveground biomass with $F_M$ treatment increased by 153.7% over the CK and did not differ significantly compared with other treatments. Under various fertilizer conditions, the aboveground biomass was highest under the $W_H$ condition. *F. coelestis* and *S. purpurea* exhibited identical trends in which decreased with increasing fertilizer application levels under various water content conditions and

gradually decreased with drought stress under various fertilizer application conditions. For both species, the aboveground biomass was highest under the $F_L$ treatment, which were respectively increased by 142.8%, 115.4%, and 107.4% and by 104.6%, 118.0%, and 108.1%, when compared with the corresponding values in the CK. Under identical fertilizer application conditions, the aboveground biomass of *F. coelestis* under the $F_L$, $F_M$, and $F_H$ treatments were higher with $W_H$ than with $W_L$ by 15.2%, 3.6%, and 51.7%, respectively, whereas the corresponding trend for *S. purpurea* was $W_H > W_L > W_M$ (Figure 3).

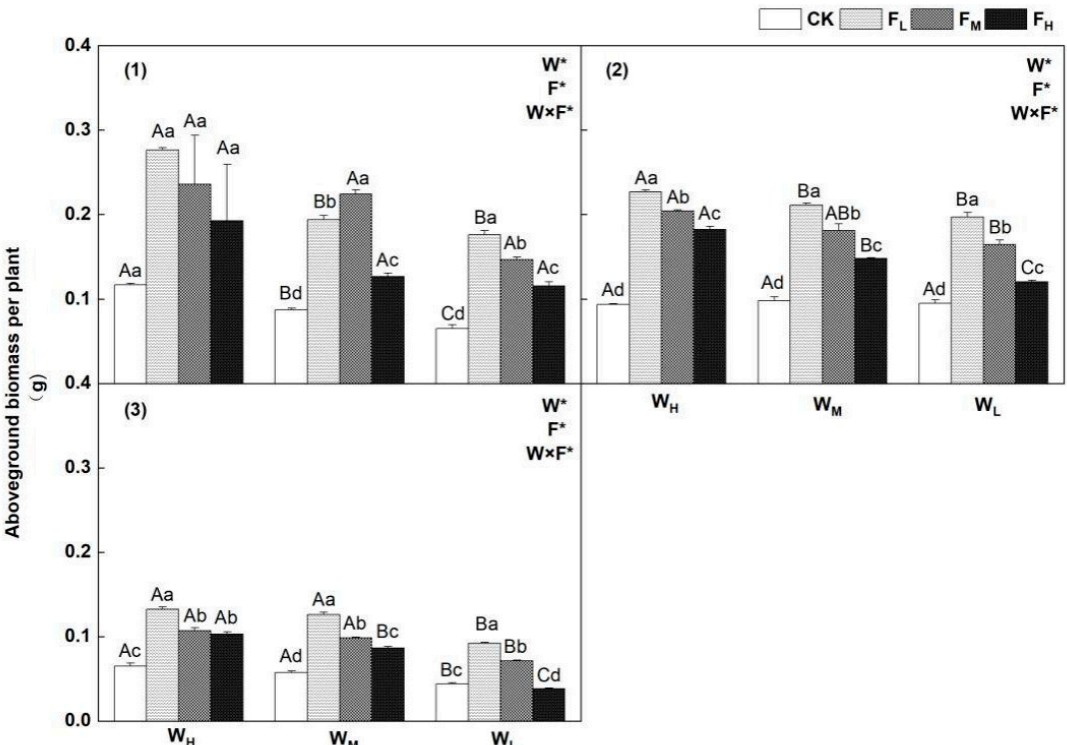

**Figure 3.** Changes in aboveground biomass values of test plant species under different water–fertilizer treatments. Numbers (**1**)−(**3**) in brackets indicate *P. crymophila*, *F. coelestis*, and *S. purpurea*, respectively. W, F, and W×F indicate the results of ANOVAs ($p < 0.05$) for the effects of water, fertilizer, and their interactions. Asterisk (*) and ns indicate significant difference and non-significant. The uppercase letters and lowercase letters represent the difference between water and fertilizer treatments.

The effects of water, fertilizer, and their interactions on the root weight were significantly different for *P. crymophila* and *S. purpurea*. Water and fertilizer significantly affected the root weight for *F. coelestis*, whereas the effects of water−fertilizer interactions were not significant ($p < 0.05$).

Differences also existed in the root weight between all three species under different treatments. Under the $W_H$ condition, the root weight for *P. crymophila* and *F. coelestis* decreased in the following order in terms of the fertilizer application level: $F_H > F_M > F_L > CK$, with the values of both species under the $F_H$ treatment being 88.7% and 45.2% higher than that of the CK, respectively. Under the $W_M$ and $W_L$ conditions, *P. crymophila* exhibited a progressive decrease in root weight as the fertilizer application level increased (except for the control treatment); the root weight under the $W_M F_L$ treatment increased by 69.4% compared with that of the CK. For *F. coelestis,* the highest values were obtained under the $F_H$ condition, which were 14.6% and 25.5% higher, respectively, compared with those of the control group. For *S. purpurea,* the highest root weight value was attained under the $W_H F_L$ conditions, which was 56.3% higher than that of the control group. Under the CK and $F_L$ conditions, the root weight was highest under the $W_L$ treatment for *P. crymophila* and *F. coelestis*. Under the $F_H$ and $F_M$ conditions, the root weights of *P. crymophila* under the $W_H$ treatment was higher than those

of other water content levels, with the values being 234.2% and 98.2% higher, respectively, than that of the $W_L$ group. The root weight of *F. coelestis* was highest under the $W_L$ treatment, but the differences compared with other treatments were relatively small. Under identical fertilizer application levels, the root weight of *S. purpurea* was highest under the $W_H$ treatment and differed significantly from those of other treatments ($p < 0.05$; Figure 4).

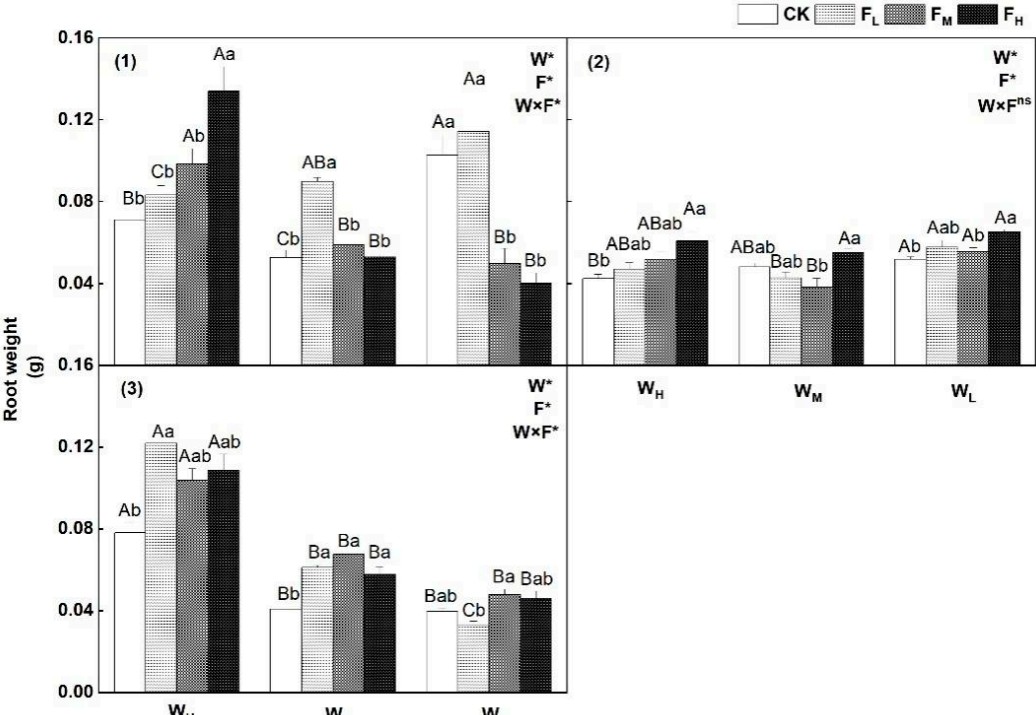

**Figure 4.** Changes in root weight values of test plant species under different water–fertilizer treatments. Numbers (**1**)−(**3**) in brackets indicate *P. crymophila*, *F. coelestis*, and *S. purpurea*, respectively. W, F, and W×F indicate the results of ANOVAs ($p < 0.05$) for the effects of water, fertilizer, and their interactions. Asterisk (*) and ns indicate significant difference and non-significant. The uppercase letters and lowercase letters represent the difference between water and fertilizer treatments.

Water and fertilizer exerted effects on the $G_s$ of *P. crymophila*, *F. coelestis* and *S. purpurea*. Different water and fertilizer conditions led to significant differences in the $G_s$ values of *P. crymophila*, whereas the differences in the effects of water–fertilizer interactions were not significant. For *F. coelestis* and *S. purpurea*, the effects of water, fertilizer, and their interactions on $G_s$ were significantly different ($p < 0.05$).

For *P. crymophila*, under the $W_H$ and $W_L$ conditions, the $G_s$ values obtained after the $F_H$ treatment were higher than after the other treatments, with the $G_s$ values being 55.6% and 78.8% higher than that of the CK, respectively. For *F. coelestis*, under the $W_H$ and $W_L$ conditions, the daily dynamic changes in the $G_s$ values exhibited a double-peak pattern under the $F_M$ treatment. The peak values occurred from 11:00−13:00 and 15:00−17:00 and were higher than those of the other treatments, with the values being 39.4% and 43.5% higher than that of the CK, respectively. Under other treatments, the $G_s$ values exhibited a single-peak pattern. For *S. purpurea*, the $G_s$ values exhibited quite similar trends under different treatments. Under the $W_H$ and $W_L$ conditions, the peak values under the $F_L$ treatment occurred from 11:00−13:00 and 15:00−17:00 and were higher than those found with other treatments. Under the $W_H$ condition, the $G_s$ value obtained under the $F_L$ treatment was 57.7% higher than that found under the CK. Under $W_M$ treatment, fertilizer application had a smaller effect on the $G_s$ values.

The effects of different water−fertilizer treatments on daily average $G_s$ values differed significantly following the various treatments. In *P. crymophila* and *F. coelestis*, under the

$W_H$ condition, the daily average $G_s$ value observed under the $F_H$ treatment was higher than those observed under other treatments. Under the $W_M$ and $W_L$ conditions, the trends in the daily average $G_s$ values of *P. crymophila* were identical, with the daily average $G_s$ being highest after $F_H$ treatment and 24% and 66% higher than that of the CK, respectively. The daily average $G_s$ value attained under the $W_HF_M$ condition was 92.2% and 122.0% higher than those attained after the $W_MF_M$ and $W_LF_M$ treatments, respectively. For *F. coelestis*, the highest daily average $G_s$ value under the $F_M$ condition was 47.8% and 29.0% higher than that of the CK, respectively. Under identical fertilizer application levels, the daily average $G_s$ values of *P. crymophila* and *F. coelestis* decreased in the following order of water content levels: $W_H > W_M > W_L$ and $W_M > W_H > W_L$, respectively. In *S. purpurea*, the $G_s$ values exhibited identical trends under the $W_H$ and $W_L$ conditions. Under the $W_H$ condition, the highest daily average $G_s$ value observed after the $F_L$ treatment was 34.3% higher than that of the CK. Under identical fertilizer application levels, the daily average $G_s$ values observed after the $F_L$ treatment were in the following order: $W_H > W_L > W_M$, with these differences being statistically significant. Under the $F_H$ condition, the daily average $G_s$ values found after the $W_M$ treatment were 32.4% and 36.1% higher than those observed after the $W_H$ and $W_L$ treatments, respectively ($p < 0.05$; Figure 5).

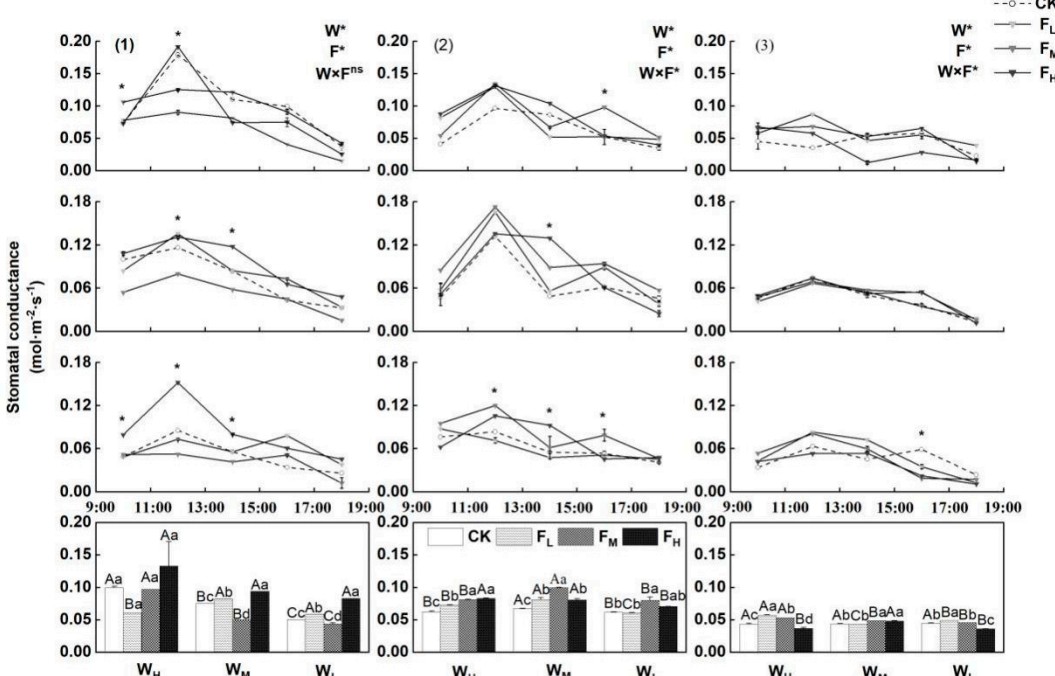

**Figure 5.** Changes in $G_s$ values of test plant species under different water–fertilizer treatments. Numbers (**1**)−(**3**) in brackets indicate *P. crymophila*, *F. coelestis*, and *S. purpurea*, respectively. W, F, and W×F indicate the results of ANOVAs ($p < 0.05$) for the effects of water, fertilizer, and their interactions. Asterisk (*) and ns indicate significant difference and non-significant. The uppercase letters and lowercase letters represent the difference between water and fertilizer treatments.

Water−fertilizer treatments exerted different effects on the stomatal frequency of the three species. The effects of water, fertilizer, and their interactions on the stomatal frequency were significantly different between *P. crymophila* and *S. purpurea*. For *F. coelestis*, the effects of water and fertilizer were significantly different ($p < 0.05$), whereas the effects of water–fertilizer interactions were not significant. In *P. crymophila* and *F. coelestis*, under the $W_H$ condition, the stomatal frequencies found under the $F_H$ treatment were 81.4% and 18.7% higher than those of the CK, respectively. Under conditions of identical fertilizer application levels, the stomatal frequency was highest under the $W_L$ treatment (versus that of the CK) but highest under the $W_H$ treatment under all other fertilizer application

conditions. For *F. coelestis*, under the $W_L$ condition, the stomatal frequency found after the $F_L$ treatment was 15.8% higher than that found in the CK. Under identical fertilizer application level conditions, the stomatal frequency values generally decreased in the following order in terms of water content levels: $W_M > W_H > W_L$. For *S. purpurea*, under the $W_H$ and $W_M$ conditions, the stomatal frequency was highest after the $F_M$ treatment. Under the $W_L$ condition, the stomatal frequency found after the $F_H$ treatment was 42.0% higher than that of the CK. Under conditions of identical fertilizer application levels, the stomatal frequency was always highest after $W_H$ treatment (Figure 6).

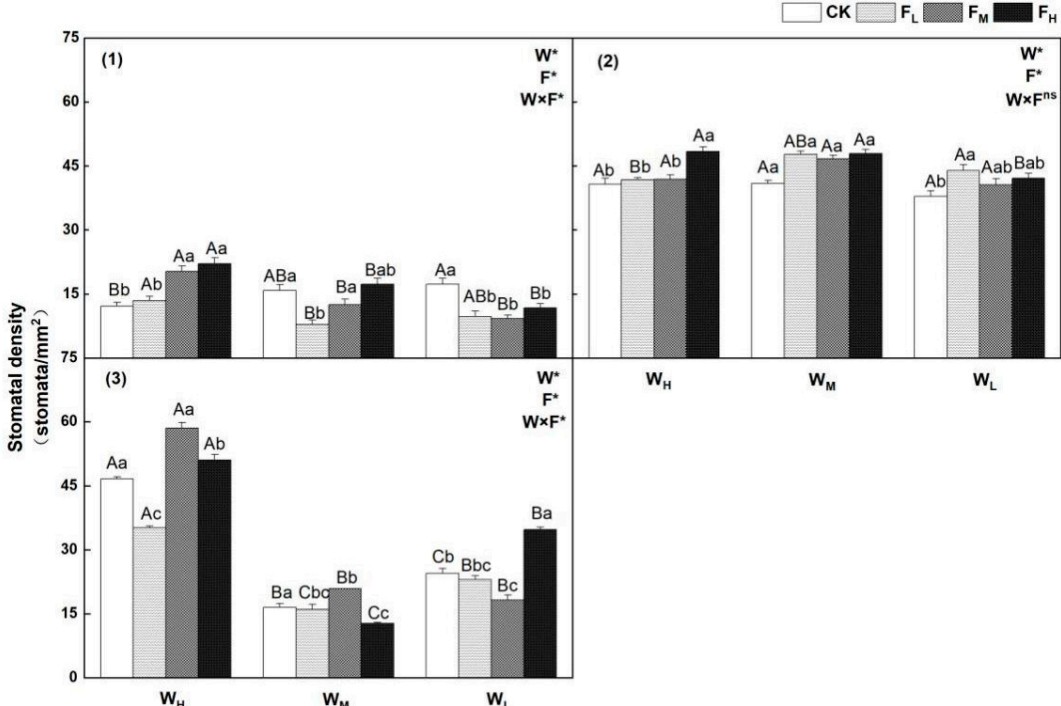

**Figure 6.** Changes in stomatal frequency values of test plant species under different water–fertilizer treatments. Numbers (**1**)−(**3**) in brackets indicate *P. crymophila*, *F. coelestis*, and *S. purpurea*, respectively. W, F, and W×F indicate the results of ANOVAs ($p < 0.05$) for the effects of water, fertilizer, and their interactions. Asterisk (*) and ns indicate significant difference and non-significant. The uppercase letters and lowercase letters represent the difference between water and fertilizer treatments.

Different water−fertilizer treatments affected the $T_r$ values of *P. crymophila*, *F. coelestis*, and *S. purpurea*. Water, fertilizer, and their interactions significantly affected the $T_r$ values of *P. crymophila* and *F. coelestis*. For *S. purpurea*, water and water–fertilizer interactions caused significant differences in the $T_r$ values ($p < 0.05$), whereas different fertilizer conditions did not significant affect the $T_r$ values.

Under the $W_H$ and $W_L$ conditions, the daily dynamic changes in the $T_r$ values of *P. crymophila* and *F. coelestis* generally exhibited a single-peak pattern. Under the $W_L F_H$ condition, the peak $T_r$ value that occurred from 11:00−13:00 was significantly higher than those of the other treatments, which showed 55.4% and 59.6% higher $T_r$ values than the CK, respectively. At the $W_M$ water content level, the $T_r$ value of *F. coelestis* showed a double-peak pattern, and the $T_r$ values observed with other treatments were higher than that of the CK. Under different water content levels, the $T_r$ of *S. purpurea* generally exhibited a single-peak pattern, with the respective peak values occurring from 11:00−13:00, 13:00−15:00, or 15:00−17:00. The Tr value was highest after the $W_L F_L$ treatment, with the value being 39.4% higher than that of the CK ($p < 0.05$). The effects of fertilizer treatment on the daily average $T_r$ value differed across different water-treatment conditions. For *P. crymophila*, under the $W_M$ and $W_L$ conditions, the daily average $T_r$ values observed under the $F_H$ treatment were higher than those of the other treatments, with the $T_r$ values increasing by 33.7%

and 30.9%, respectively, compared with that of the CK. Under the $W_H$ condition, the daily average $T_r$ values decreased in the following order in terms of the fertilizer application level: $F_M > F_H > CK > F_L$, with the $T_r$ values of the CK and $F_M$ groups under the $W_H$ and $W_M$ conditions being significantly different from those under the $W_L$ condition. Under the $F_L$ and $F_H$ conditions, the daily average $T_r$ value found after the $W_M$ treatment differed from those after the $W_H$ and $W_L$ treatments. For *F. coelestis*, the daily average $T_r$ increased with an increasing fertilizer application level under the $W_H$ and $W_L$ conditions, with the value observed after the $W_H F_H$ treatment being 26.1% higher than that under the CK. Under the $W_M$ condition, the daily average $T_r$ value was highest after the $F_M$ treatment and differed significantly from the values found after the other treatments. For *S. purpurea*, the effect on the daily average $T_r$ was higher for the CK than for the other treatments under the $W_H$ and $W_L$ conditions. The daily average Tr was highest after the $F_H$ treatment under the $W_M$ conditions, being 17.4% higher than the CK ($p < 0.05$). Under the CK, $F_L$, and $F_M$ conditions, the daily average $T_r$ value decreased in the following order in terms of the water content level: $W_H > W_L > W_M$. Under the $F_H$ condition, the daily average $T_r$ value was highest in the $W_M$ group (Figure 7).

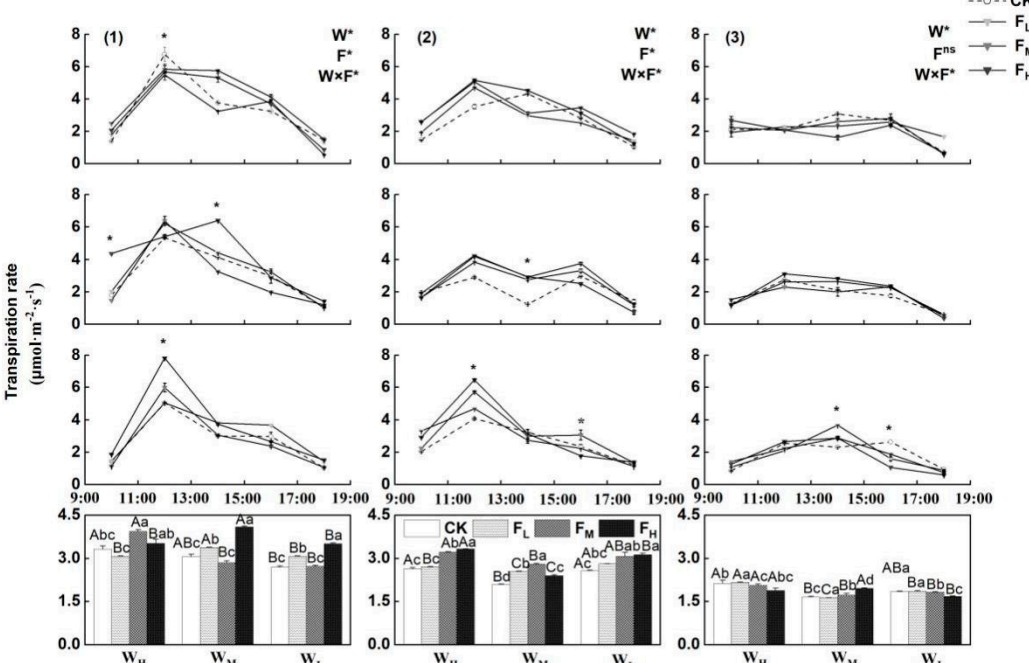

**Figure 7.** Changes in $T_r$ values of test plant species under different water–fertilizer treatments. Numbers (**1**)−(**3**) in brackets indicate *P. crymophila*, *F. coelestis*, and *S. purpurea*, respectively. W, F, and W×F indicate the results of ANOVAs ($p < 0.05$) for the effects of water, fertilizer, and their interactions. Asterisk (*) and ns indicate significant difference and non-significant. The uppercase letters and lowercase letters represent the difference between water and fertilizer treatments. *F. coelestis*, and *S. purpurea* were significantly different ($p < 0.05$).

Water consumption of the three species decreased when the water content level decreased. With an increase in the fertilizer application level, water consumption increased under the $W_H$ treatment and decreased under the $W_M$ and $W_L$ treatments. The water consumption levels of *P. crymophila*, *F. coelestis*, and *S. purpurea* were 200.2 g/pot, 142.3 g/pot, and 169.9 g/pot, respectively, with the water consumption of *P. crymophila* being the highest of the three species.

In *P. crymophila*, under the $W_H$ condition, water consumption after the $F_H$ treatment was 49.2% higher than that of the CK. If water consumption under the $W_M$ treatment was set to 100%, then water consumption under the CK at the $W_H$ and $W_L$ water content levels would be 78.2% and 60.9%, respectively. The corresponding water consumption values under the $F_L$, $F_M$, and $F_H$ treatments would be 102.7% and 53.1%, 146.2% and 66.2%, and 178.1% and 31.0%,

respectively. For *F. coelestis* and *S. purpurea*, under the $W_H$ condition, water consumptions after the $F_H$ treatment were 49.0% and 25.9% higher than that under the CK, respectively. Under an identical fertilizer application level, the water consumption of *F. coelestis* significantly decreased ($p < 0.05$) with decreasing water content levels (Figure 8).

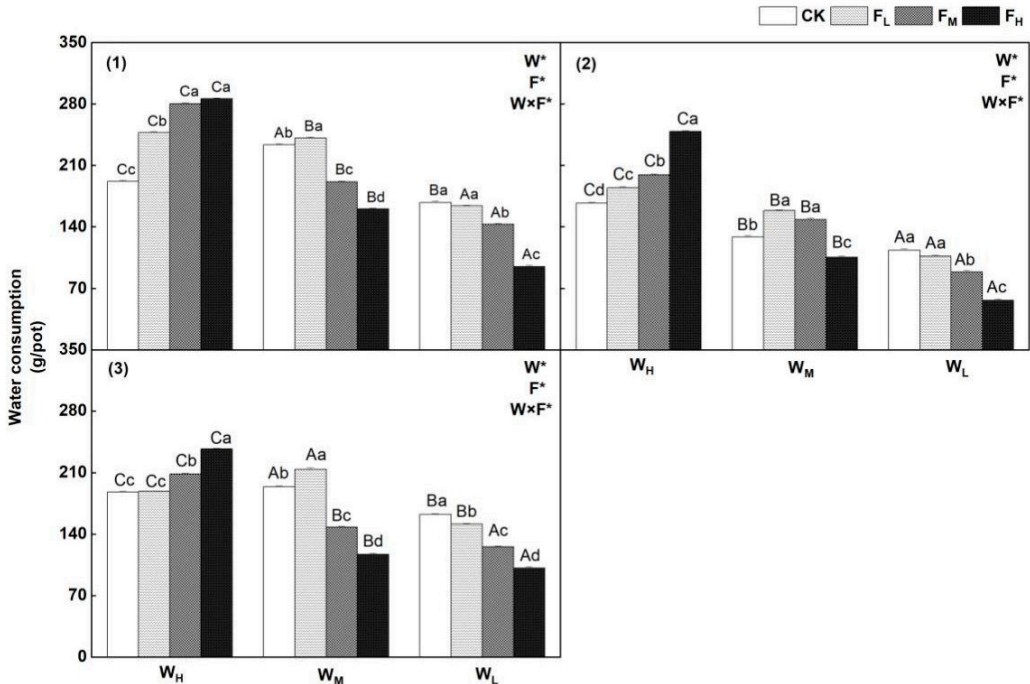

**Figure 8.** Changes in water consumption values of test plant species under different water–fertilizer treatments. Numbers (**1**)−(**3**) in brackets indicate *P. crymophila*, *F. coelestis*, and *S. purpurea*, respectively. W, F, and W×F indicate the results of ANOVAs ($p < 0.05$) for the effects of water, fertilizer, and their interactions. Asterisk (*) and ns indicate significant difference and non-significant. The uppercase letters and lowercase letters represent the difference between water and fertilizer treatments.

Water–fertilizer treatments affected the WUE of *P. crymophila*, *F. coelestis*, and *S. purpurea*. The effects of water, fertilizer, and their interactions on the WUE were significant for *S. purpurea* and *F. coelestis* ($p < 0.05$). For *P. crymophila*, the effects of water and water–fertilizer interactions on the WUE were significant, whereas the effects of different fertilizer levels were not significant.

For *P. crymophila*, under the $W_H$ and $W_M$ conditions, the WUE exhibited a double-peak pattern under the $F_H$ treatment and a single-peak pattern under all other treatments. Under the $W_H F_L$ treatment, the peak WUE value from 17:00−19:00 was higher than that of other treatments and was 46.1% higher than that of the CK. For *F. coelestis*, the daily dynamic trends of the WUE were generally identical under the $W_H$ and $W_M$ conditions, indicating that variations in the fertilizer application had a small effect on the WUE. Under the $W_L F_H$ treatment, the peak value occurring from 15:00−17:00 was 85.4% higher than that under the CK. For *S. purpurea*, under the $W_H$ and $W_L$ conditions, the WUE exhibited a generally identical double-peak pattern, with peak values occurring from 11:00−13:00 and 15:00−17:00. Water–fertilizer treatments had a smaller effect under the $W_M$ condition. Differences were found in the effects of different water–fertilizer treatments on the daily average WUE. For *P. crymophila*, under the $W_H$ condition, the daily average WUE with $F_H$ treatment was 34.5% higher than that with $F_M$ treatment; under the $W_L$ condition, the effects of various treatments were relatively small; under the $F_H$ condition, the WUE observed with $W_H$ treatment was 47.3% higher than that found with $W_M$ treatment. For *F. coelestis*, the daily average WUE generally increased with increasing fertilizer application levels. The highest daily average WUE was attained under the $W_M F_H$ treatment, which

was 47.2% higher than that under the CK. Under an identical fertilizer application level, the daily average WUE values decreased in the following order in terms of the water content level: $W_M > W_L > W_H$. For *S. purpurea*, under the $W_L$ condition, the daily average WUE increased with increasing fertilizer application levels, with the WUE value being 31.1% higher under the $F_H$ treatment than under the CK ($p < 0.05$). Under the CK and $F_L$ conditions, the daily average WUE was highest under the $W_M$ treatment. Under the $F_H$ and $F_M$ conditions, the daily average WUE decreased in the following order in terms of the water content level: $W_L > W_M > W_H$ (Figure 9).

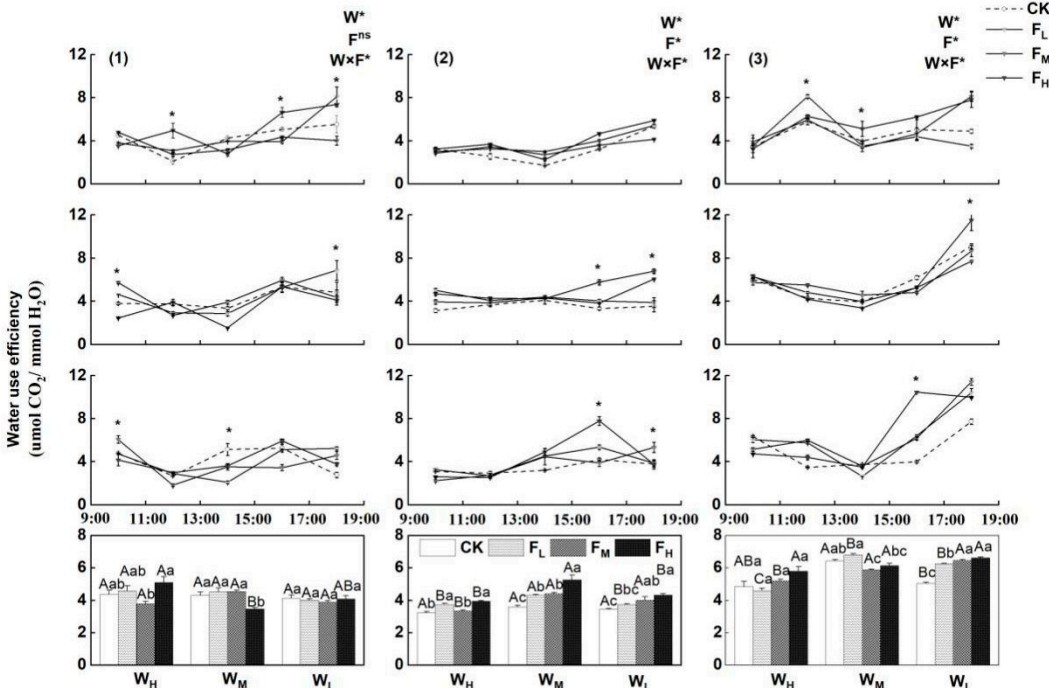

**Figure 9.** Changes in WUE values of test plant species under different water–fertilizer treatments. Numbers (**1**)−(**3**) in brackets indicate *P. crymophila*, *F. coelestis*, and *S. purpurea*, respectively. W, F, and W×F indicate the results of ANOVAs ($p < 0.05$) for the effects of water, fertilizer, and their interactions. Asterisk (*) and ns indicate significant difference and non-significant. The uppercase letters and lowercase letters represent the difference between water and fertilizer treatments.

The stomatal sizes of the three species were the largest under $W_H$. Specifically, stomatal sizes decreased with the stress of water deficiency for all *P. crymophila* plants under fertilizer treatments, *F. coelestis* under $F_M$ and $F_H$, and *S. purpurea* under the CK condition. The stomatal size of *P. crymophila* increased with the application rate of fertilizer, whereas *F. coelestis* and *S. pupurea* had little effect (Figure 10).

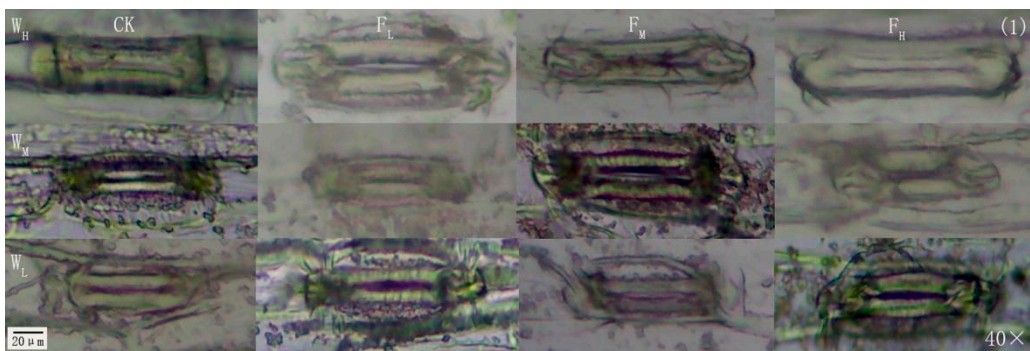

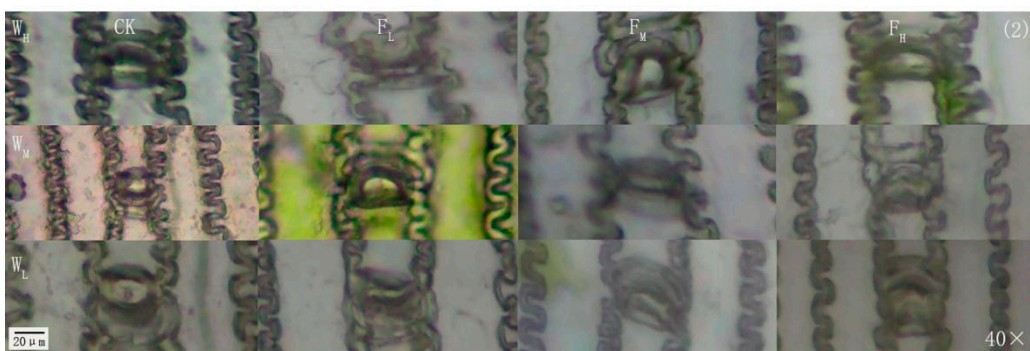

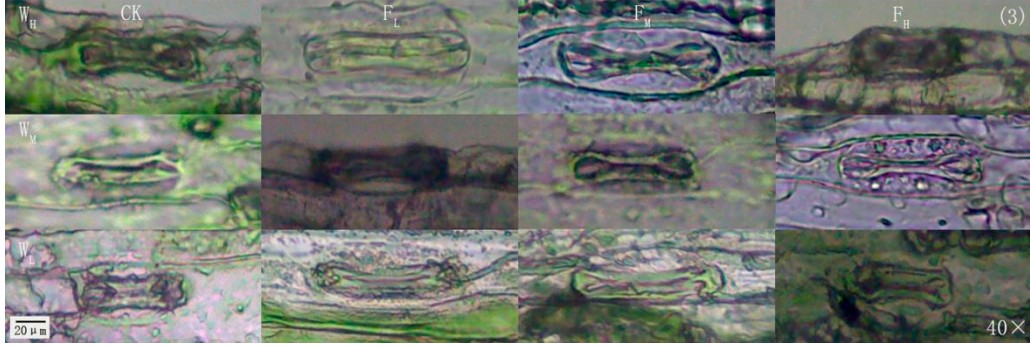

**Figure 10.** Changes in stomatal size values of test plant species under different water–fertilizer treatments. Numbers (**1**)−(**3**) in brackets indicate *P. crymophila*, *F. coelestis*, and *S. purpurea,* respectively.

## 4. Discussion

### 4.1. Effects of Different Water−Fertilizer Treatments on the Biomass

Different water–fertilizer conditions effectively increased the aboveground biomass, with the biomass of *P. crymophila* being higher than those of the other two species. The leaf number and leaf thickness of the three species generally decreased with reductions in the water content level, consistent with the trend exhibited with the aboveground biomass. Therefore, it is evident that the increase in aboveground biomass was limited by water conditions. This conclusion is consistent with the results reported by Zhang [28] in a study on *Puelia sinese* Roxb conducted on the Sichuan Plateau, which may be ascribed to the fact that plants invest less in stems and leaves under conditions of drought [29,30]. For *F. coelestis* and *S. purpurea*, appropriate fertilizer application under drought stress favored an increase in the aboveground biomass. However, an increase in the fertilizer application level increased the osmotic potential of water, leading to a reduced nutrient absorption ability of plants that ultimately resulted in a lower plant biomass. The root weight differed after the $W_H$ and $W_M$ treatments. The root weight for all *S. purpurea* plants with added fertilizer was higher than that of the plants under the CK, which demonstrates that fertilizer application enabled better nutrient absorption and transport to other plant parts [31]. For

*F. coelestis*, under the $W_M$ condition, only the root weight observed after $F_H$ treatment was higher than that after the CK. Under the $W_L$ condition, the root weight values of all treatment groups were higher than those of the CK.

These findings indicate that under drought conditions, the assimilated materials generated in the aboveground parts of plants were mainly transported to the belowground parts, thereby promoting root growth. Consequently, water absorption by roots was enhanced, which benefited the adaptation of plants to drought stress. Compared with the other two species, *P. crymophila* exhibited greater differences after the $W_M$ and $W_L$ treatments. Under the $W_M$ condition, fertilizer application led to decreasing trends in aboveground biomass and in the root weight, which further exemplified the existence of a parallel relationship, i.e., biomass accumulation and allocation are characteristics of environmental adaptation in plants [32]. Under the $W_L$ treatment, drought stress may have caused a decrease in antioxidant enzyme activities in the plants [33], which can affect the metabolic ability. Consequently, most assimilated materials in the aboveground parts were not transported to the belowground parts, which lowered the biomass.

*4.2. Effects of Different Water−Fertilizer Treatments on the $P_n$, Leaf Number and Leaf Thickness*

Photosynthesis in leaves is a basic physiological process of plants [34], and both water and fertilizer significantly affect the physiological characteristics of growth, development, and photosynthesis in plants [35]. Photosynthesis is affected through the effects exerted on multiple factors such as the stomata [36] and photosynthetic pigment content [37]. Such changes in turn affect the leaf number, leaf thickness, and biomass accumulation of plants, which are indirect manifestations of the $P_n$ value. An increase in the leaf number leads to an increased total plant area that is capable of receiving light energy; with the increase in leaf thickness, stomata and palisade cell layers become common on both leaf sides, thereby broadening the zone of light and $CO_2$ inside the leaf and increasing the amount of photosynthesis per unit leaf area [38]. Under drought stress, plant roots absorb less water and photosynthetic activity is reduced. However, these changes can be regulated to a certain extent through changes in soil nutrients [39]. Our results indicate that different water–fertilizer-coupling combinations exerted significant effects on the $P_n$ values of all three species, which agrees with previous findings [40]. The $P_n$ of *P. crymophila* was more sensitive to changes in water−fertilizer treatments than the other two species and exhibited a larger range in peak values with various treatments under the $W_H$ and $W_L$ conditions. This may be explained by the fact that suitable water content conditions contribute to an increase in the $G_s$ value, which in turn causes an increase in the $P_n$ value [41]. With aggravated water deficiency stress, the original double-peak pattern exhibited by the daily dynamic changes of $P_n$ of *F. coelestis* changed to a single-peak pattern. This outcome may be related to the decrease in $G_s$ under drought conditions, which increased photosynthesis and alleviated midday depression. The daily average $P_n$ value was highest under the $W_H F_H$ treatment for *P. crymophila* and *F. coelestis* and under the $W_H F_L$ treatment for *S. purpurea*. These findings may reflect the response of *S. purpurea* under adverse habitat conditions.

Our findings further demonstrate differences in the adaptability of the three plant species under different water and fertilizer conditions. Reasonable water and fertilizer conditions can improve the water contents of crops, enhance the ability of plants to regulate the osmotic potential and stomata, and increase the photosynthesis rate [42,43]. The leaf number and leaf thickness of the three species were largest under the $W_H F_L$ treatment and decreased in all fertilizer treatment groups with lower water and fertilizer contents. These results demonstrate that photosynthesis in plant leaves is not clearly associated with the leaf number but is related to the chloroplast number and photosynthetic rate. Both *P. crymophila* and *F. coelestis* possess more tillers and leaves and shading is largely prevalent and reduces the number of plant cells and causes leaf thinning during leaf growth [44]. Thinner leaves contain thinner palisade tissue and fewer chloroplasts, which hinders $CO_2$ transport and dissolution [45]. Therefore, thinner leaves do not have strong photosynthetic and biomass accumulation abilities [46]. Compared with the two aforementioned species,

*S. purpurea* exhibited significantly fewer leaves, less shading, and thicker leaves, which increased the photosynthesis efficiency [47].

*4.3. Effects of Different Water−Fertilizer Treatments on the $G_s$, Stomatal Frequency, and Stomatal Size*

Stomata are channels through which water and $CO_2$ within plants are exchanged with the external environment. The stomatal size, stomatal frequency, and $G_s$ significantly affect plant transpiration [48]. In general, a decreased stomatal aperture and an increased stomatal density are characteristic plant responses to drought environments [49]. However, the response of the stomatal density to drought is not merely related to the plant type but is also related to the degree of drought stress [50]. The adaptation of stomata to drought is usually manifested as a decreased stomatal size and an increased stomatal density, with consistent lengths and widths or an increased size and sparsity. Our results indicate that the daily average $G_s$ of the three species under different irrigation and fertilizer application conditions were higher than the corresponding CK values. Under identical fertilizer application levels, the $G_s$ decreased with increasing water deficiency, which agrees with previous findings [51]. Stomatal sizes decreased with water deficiency stress for all *P. crymophila* plants subjected to fertilizer treatment, *F. coelestis* plants subjected to $F_M$ and $F_H$ treatment, and *S. purpurea* plants under the CK condition (Figure 10). As stomatal growth and development were limited by the water content, *P. crymophila* and *S. purpurea* plants under CK showed decreasing $G_s$ and $T_r$ values with decreasing stomatal sizes under drought stress, which affected photosynthesis and provided an adaptive mechanism under drought conditions. Similar patterns were observed previously with *Arabidopsis thaliana* [52]. However, under the $F_M$ and $F_H$ conditions, the results found for *F. coelestis* were inconsistent with those found for the other two species. Under drought conditions, the stomatal size decreased but the $T_r$ increased, leading to a decrease in WUE. Therefore, excessive fertilizer application under drought conditions did not attenuate the detrimental effects of water deficiency.

For *P. crymophila* and *F. coelestis*, the daily average $G_s$ and stomatal frequency were highest under the $W_H F_H$ treatment. A higher $G_s$ promotes the greater $CO_2$ uptake into cells, which leads to higher photosynthetic activity. Under the $W_H$ condition, the $G_s$ correlated positively with the stomatal frequency. An increased stomatal frequency led to an increased $G_s$, which facilitated water and $CO_2$ exchange between plant leaves and the external environment, thereby increasing the $P_n$. Under the $W_L$ condition, the daily average $G_s$ was generally low, which was a common adaptation characteristic of both species to drought conditions. The stomatal frequency was not related to the $G_s$, which may reflect the combined effects of multiple factors such as environmental conditions. For *P. crymophila*, the $G_s$ exhibited a single-peak pattern under different water–fertilizer treatments, with the peak value decreasing with increasing water deficiency stress. These results indicate that soil water deficiency led to decreased water absorption by roots, which lowered the leaf water contents and led to leaf guard cell shrinkage, ultimately resulting in a decreased $G_s$ [53]. Under identical water content conditions, the daily average $G_s$ was highest under the $F_H$ treatment, demonstrating that nitrogen could regulate the $G_s$, consistent with findings reported by Li et al. [54]. Our findings also showed that the stomatal frequency of *P. crymophila* decreased with increasing water deficiency stress under all fertilizer treatments (except for CK), which agrees with the $G_s$ results. These data indicate that the *P. crymophila* leaves exhibited some trait plasticity when adapting to drought stress [55].

Fertilizer application under drought stress led to a significant decrease in the stomatal frequency. This finding may reflect the low stomatal density caused by nutrient limitations imposed by nitrogen on early guard cell development during drought stress [56,57]. The $G_s$ of *F. coelestis* was highest under the $W_M F_M$ treatment, which demonstrates that applying an appropriate amount of nitrogen fertilizer under suitable water conditions can contribute to an increased $G_s$. The stomatal frequency generally increased with added fertilizer under identical irrigation conditions. Although the stomatal frequency was highest under the $W_H F_H$ treatment, it was also close to the values attained with various fertilizer treatments under the $W_M$ condition, which may relate to other factors, such as the environment, temperature,

or light intensity. In general, the $G_s$ of *S. purpurea* was significantly lower than those of *P. crymophila* and *F. coelestis*, which may reflect differences in the inherent characteristics of the different species. The rangeability of $G_s$ under the $W_H$ and $W_L$ conditions was greater than that under the $W_M$ condition, indicating that fertilizer applications under $W_H$ and $W_L$ exerted a greater effect on $G_s$. The daily average $G_s$ decreased with increasing fertilizer application levels under the $W_H$ and $W_L$ conditions and was highest under the $W_H F_L$ treatment. These results show that excessive fertilizer application under adequate water content or drought stress limited the increase in $G_s$ of *S. purpurea*, which is consistent with previous results reported by Maggard et al. [58]. The relationship between $G_s$ and $P_n$ in *S. purpurea* was also consistent with the corresponding relationship in *P. crymophila*. The stomatal frequency also differed depending on the plant type and degree of stress [55]. The stomatal frequency of *S. purpurea* exhibited a V−shaped trend with decreasing water content. Under the three different water conditions, excessive fertilizer application favored an increase in the stomatal frequency, with the value being highest under the $W_H F_M$ treatment.

*4.4. Effects of Different Water−Fertilizer Treatments on the $T_r$, Water Consumption and WUE*

The soil water content is an important indicator of water and fertilizer use by plants. Soil water is mainly consumed by evaporation, transpiration, and plant absorption. $T_r$ is an essential physiological function of plant leaves, and its relationship with the environment is key for plants to adapt to drought conditions. Our results indicated that the $T_r$ and water consumption values were higher in *P. crymophila* and *F. coelestis* and lower in *S. purpurea*. Under identical water content conditions, total water consumption differed significantly with different fertilizer application levels. In the presence of sufficient water, the water consumption of all three species increased with increasing fertilizer application levels, but the opposite trend was observed under conditions of appropriate water content and drought stress. For *P. crymophila*, the $T_r$ exhibited a single-peak pattern, and the daily average value was highest under an appropriate water content. These findings indicate that both excessively low and excessively high soil water contents caused decreased transpiration. An excessively low soil water content causes plant roots to generate signals that are subsequently transmitted to leaf stomata, which promotes decreased $G_s$ and transpiration levels. When the soil water content is excessively high, the air permeability of soil becomes poorer, and the decreased root respiration reduces the water absorption capability of roots, thereby reducing transpiration [59]. Water consumption and the WUE were highest under the $W_H F_H$ treatment, indicating that fertilizer application in the presence of an appropriate water content can enhance the ability of plants to regulate osmosis. In particular, the application of nitrogen fertilizer significantly reduced the $T_r$ and increased the WUE [60].

The peak $T_r$ value of *F. coelestis* changed significantly under different water content conditions. As the water content decreased, the $T_r$ trend generally exhibited a double peak−single peak−double peak transition. The daily average $T_r$ values under the $W_H$ and $W_L$ conditions were higher than those under the $W_M$ condition, which did not correlate positively with changes in the $G_s$, in contrast to the findings of Liu [61]. Our findings may have been due to changes in the atmospheric evaporative demand and soil water content, leading to changes in leaf water potential and a lack of effective regulation of the plant stomata. A decreased soil water content promotes abscisic acid synthesis in plant roots, which is transported to leaves via the water transport route and induces stomatal closure [62]. The highest daily average $T_r$ value was attained under the $W_H F_H$ treatment, which is consistent with the highest water consumption value observed under the same treatment. These data show that water consumption and transpiration were highest for *F. coelestis* with a sufficient water content, which led to an increase in the $P_n$ value.

For *S. purpurea*, the rangeability of the daily dynamic changes of the $T_r$ value was smaller under identical water content conditions. The daily average $T_r$ value was highest under the $W_H F_L$ treatment, which is consistent with the highest $P_n$ and $G_s$ values being observed under the same condition. Such a result indicates that the transpiration rate was largely determined by the state of stomatal activity and that fluctuations in stomatal activity

were caused by plant adaptation to the water content state [63], which is consistent with the findings of Qiao [64]. The daily average WUE under the $W_L$ condition was generally higher than that under the $W_H$ condition, which is consistent with the results reported by Jaleel [65]. Given that the $G_s$ value decreased in plants under water deficiency stress, which caused a decreased transpiration rate and increased WUE, our results demonstrate the adaptability of plants to adversity. However, a higher WUE under drought conditions is not favorable for increased production, as an increased WUE contributed to a reduction in yield. Drought stress severely inhibits plant growth and reduces water consumption, thereby causing a relative increase in the WUE [66].

**5. Conclusions**

Drought and nutrient deficiency has become a serious ecological problems in the alpine grassland of the Qinghai−Tibet Plateau. Our results showed that drought stress reduced the biomass of test plants via limiting photosynthesis and water use, but the influence could be mitigated by nitrogen addition through a complementary effect owing more on photosynthesis than to water−related traits. The test plants displayed different tolerances to environmental stress in biomass production and physiological sensitivity. Specifically, both above- and belowground *P. crymophila* and *S. purpurea* were influenced by water−fertilizer interaction. In contrast, the aboveground biomass of *F. coelestis* was mainly controlled by fertilizer, whereas the belowground biomass was not sensitive. *P. crymophila* and *S. purpurea* showed regulatory changes in both photosynthesis and water use, with less stomatal frequency and reduced stomatal size, but the latter had a lower transpiration rate and water consumption; therefore, although the WUE of *S. purpurea* was higher than that of *P. crymophila*, it resulted in a lower biomass. With regard to *F. coelestis*, it is characterized by a stable stomatal size but a higher stomatal frequency; therefore, the water−fertilizer interaction had a larger effect on water consumption than on photosynthesis. Consequently, the three tested species could effectively reduce the risk of establishment failure in ecological restoration with reasonable water−fertilizer combinations. $W_H F_H$ for *P. crymophila* and *F. coelestis* and $W_H F_L$ for *S. purpurea* are suitable for the reseeding of the Tibetan Plateau or other alpine regions. This work provides useful information for the sustainable management of alpine grasslands and understanding of the water−nutrient relationship in herbaceous plants in harsh alpine regions.

**Author Contributions:** Writing−original draft preparation, N.Z.; writing−original draft preparation, data curation, X.S. and S.H.; data curation, software, G.C., H.Z., Y.H. and J.Z.; supervision, validation, writing−review and editing, X.W. and Z.Z. All authors have read and agreed to the published version of the manuscript.

**Funding:** This work was financially supported by the National Natural Science Foundation of China (42161012); the Fund Project for Central and Local Universities in 2022 (KY2022ZY-01), and the Northwest A&F University & Tibet Agricultural and Animal Husbandry University Collaborative Fund (2452020044).

**Institutional Review Board Statement:** Not applicable.

**Data Availability Statement:** Not applicable.

**Conflicts of Interest:** The authors declare no conflict of interest.

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
