# Peer review of "N Addition Mitigates Water Stress via Different Photosynthesis and Water Traits for Three Native Plant Species in the Qinghai–Tibet Plateau"

_agriculture, doi:10.3390/agriculture12111873_

Round 1

Reviewer 1 Report

minor corrections are required. which are highlighten in attached file 

Reviewer 2 Report

1.      Write full names of  P. crymophila, F. coelestis, and  S. purpurea in abstract

2.      Write full form of CK and WUE in abstract

3.      Rewrite the keywords by avoiding the words used in the title

4.      Give brief botanical description of Poa crymophila, Festuca coelestis, and Stipa purpurea in indroduction section

5.      Replace “experimentation” with “experiment”

6.      Give full form of “WUEs” in section 2.2.4

7.      Give suitable citation for “Drought stress can causes stomatal closure and …… retards photosynthesis” line no. 45-46 author can refer-

a.       Krishna, R., Ansari, W. A., Jaiswal, D. K., Singh, A. K., Prasad, R., Verma, J. P., & Singh, M. (2021). Overexpression of AtDREB1 and BcZAT12 genes confers drought tolerance by reducing oxidative stress in double transgenic tomato (Solanum lycopersicum L.). Plant Cell Reports40(11), 2173-2190.

b.      Krishna, R., Ansari, W. A., Jaiswal, D. K., Singh, A. K., Verma, J. P., & Singh, M. (2021). Co-overexpression of AtDREB1A and BcZAT12 increases drought tolerance and fruit production in double transgenic tomato (Solanum lycopersicum) plants. Environmental and Experimental Botany184, 104396.

8.      Give a global and Tibet drought scenario

9.      The reference [7] is very old update it by a suitable recent one.

10.  There are many references which are more than twenty year old, these should be replaced with recent references except those dealing with the methodologies.

11.  The introduction section written nicely but it can be improved by emphasizing drought and ecological sustainability

12.  The method and materials section also written very nicely but author should clearly describe how the achieved the different moisture contents.

13.  Merge the notes section with figure caption in all figures.

14.  Add detailed description for figure 10

15.  All the figures are ok.

16.  The results and discussion section written nicely and the interpretation of results in the discussion section are ok.

17.  The conclusion section can be improved for better soundness.

Overall the manuscript is prepared and presented in a nice way and will help in ecological restoration therefore can be accepted for publication.
